# Trends in inequalities in avoidable hospitalisations across the COVID-19 pandemic: a cohort study of 23.5 million people in England

Mark Alan Green [ID],[1] Martin McKee [ID],[2] Jon Massey [ID],[3] Brian Mackenna,[4] Amir Mehrkar,[3] Seb Bacon,[3] John Macleod,[5] Aziz Sheikh,[6] Syed Ahmar Shah [ID],[7] The OpenSAFELY Consortium,[8] The Longitudinal Health and Wellbeing National Core Study Collaborative,[9] Srinivasa Vittal Katikireddi [ID] [10]

For numbered affiliations see end of article.

**Correspondence to**
Dr Mark Alan Green;
mark.green@liverpool.ac.uk

## ABSTRACT

**Objective** To determine whether periods of disruption were associated with increased 'avoidable' hospital admissions and wider social inequalities in England.

**Design** Observational repeated cross-sectional study.

**Setting** England (January 2019 to March 2022).

**Participants** With the approval of NHS England we used individual-level electronic health records from OpenSAFELY, which covered ~40% of general practices in England (mean monthly population size 23.5 million people).

**Primary and secondary outcome measures** We estimated crude and directly age-standardised rates for potentially preventable unplanned hospital admissions: ambulatory care sensitive conditions and urgent emergency sensitive conditions. We considered how trends in these outcomes varied by three measures of social and spatial inequality: neighbourhood socioeconomic deprivation, ethnicity and geographical region.

**Results** There were large declines in avoidable hospitalisations during the first national lockdown (March to May 2020). Trends increased post-lockdown but never reached 2019 levels. The exception to these trends was for vaccine-preventable ambulatory care sensitive admissions which remained low throughout 2020–2021. While trends were consistent by each measure of inequality, absolute levels of inequalities narrowed across levels of neighbourhood socioeconomic deprivation, Asian ethnicity (compared with white ethnicity) and geographical region (especially in northern regions).

**Conclusions** We found no evidence that periods of healthcare disruption from the COVID-19 pandemic resulted in more avoidable hospitalisations. Falling avoidable hospital admissions has coincided with declining inequalities most strongly by level of deprivation, but also for Asian ethnic groups and northern regions of England.

## INTRODUCTION

There has been public, political and media concern that the disruption in healthcare during the pandemic would increase hospital admissions among people not receiving necessary care. Visits to general practice fell markedly,[1] although this was compensated for by online consultations.[2] Elective treatments in England in 2020 fell by 3 million compared with 2019.[3] Cancer screening programmes, non-essential surgeries and diagnostic procedures were postponed or cancelled.[1 4] Waiting lists grew, delaying access to care for new treatments.[5 6] High levels of staff illness from acute COVID-19 and long COVID have disrupted care delivered.[7] These trends have been observed not only in England, but also many countries.[8 9] There has been little empirical evaluation into whether such disruption translated into more hospital admissions. We only know of one study demonstrating that people who self-reported that disruption of access to healthcare had higher odds of admission for a potentially preventable condition in the first 2½ years of the pandemic.[10]

Identifying the impacts of healthcare disruption is difficult. The journey from onset of illness to accessing care is complex and obstacles can arise at many points across the health system.[11] To simplify this issue, we borrowed the concept 'avoidable hospital

admissions' from the health systems literature, considered a proxy measure of health system performance.[12] Avoidable hospitalisations are emergency (unplanned) hospital admissions that could potentially have been prevented if individuals had received timely care within the community (and therefore may be susceptible to disruptions to care).[12 13] Reducing avoidable hospital admissions is a priority for the NHS because they are often costly and disrupt elective care.[14] Significant disruption to healthcare access, coinciding with uncontrolled infection and non-pharmaceutical interventions implemented in response (eg, people unable to see their general practice (GP) doctor or postponement to treatments) may have resulted in more avoidable hospitalisations where individuals were not able to receive the care they needed.[10]

### STRENGTHS AND LIMITATIONS OF THIS STUDY

⇒ Uses electronic health records for ~23.5 million people which are representative of England.
⇒ Examines how trends in avoidable hospitalisations vary across three measures of social and spatial inequality (deprivation, ethnic group and geographic region).
⇒ Analyses are descriptive and unable to explain what factors were driving trends.
⇒ It is uncertain whether avoidable hospitalisations are a good proxy measure for capturing the impacts of healthcare disruption.

Research has shown that avoidable hospitalisations in many countries fell in 2020 during the first COVID-19 waves and did not return to pre-pandemic levels thereafter.[15–18] Most of this research, however, stops at the end of 2020 and there is less understanding of how trends have continued since. This ignores a critical period in the response and recovery from the pandemic. 2021–2022 saw significant disruption to health systems due to COVID-19.[19] It is also plausible that the impacts of disrupted access to healthcare are not immediate and may only be observable in the medium-term (eg, delayed diagnosis or treatment leading to disease progression that results in an avoidable hospitalisation a year down the line).[10] Much of this research has been undertaken using small datasets for specific geographic contexts (eg, single cities or states) which may not be generalisable to the national level. Few studies have examined how trends in avoidable hospitalisations have varied across measures of social inequalities. This is pertinent since the COVID-19 pandemic shined a light on the fractures within English society,[20] with people of lower socioeconomic position, minoritised ethnic groups and those living in parts of northern England disproportionally affected by COVID-19.[21–23] These same communities have not just only been shown to report greater experiences of healthcare disruption,[24] but they are also those which had higher rates of avoidable hospitalisations pre-pandemic too.[12 25–27] We hypothesise that the COVID-19 pandemic may have widened social inequalities in avoidable hospitalisations.

The aim of our study is to determine whether periods of healthcare disruption were associated with increased 'avoidable' hospital admissions and wider social inequalities in England.

## METHODS
### Data
The primary data source used for the analysis was Open-SAFELY-TPP. OpenSAFELY is an open-source secure health data analysis platform providing access to primary care records, linked to secondary care and mortality records, held by the two largest electronic health record providers for NHS England—Egton Medical Information Systems (EMIS) and The Phoenix Partnership (TPP) which cover ~58 million registered patients. In this study, we had access to data from TPP which covers ~40% of general practices in England. The data are broadly representative of England by age, sex, ethnicity, small area socioeconomic deprivation and cause of death.[28]

Data were accessed on 20 May 2022. The study period for our analysis was 1 January 2019 to 31 March 2022. While 2019 only provides 1 year's worth of pre-pandemic data, overall trends prior to 2019 were relatively flat.[29] We discounted all data in April and May 2022 to minimise under-counting of events due to possible delays in reporting of clinical information relating to admissions. Total counts per month for all outcomes were rounded to their nearest 5 to minimise risks of patient identification where low counts were evident (a condition in the standard data agreement). As this only adjusted monthly counts by 1–3 units, the impact was relatively minimal. We redacted counts <10 to meet statistical disclosure requirements. All analytical scripts, processed data and outputs are openly available in online (https://github.com/opensafely/avoidable_hospitalisations_trends).

### Outcomes
Avoidable hospitalisations were defined using five measures commonly used by NHS England.[12] Each was defined using ICD-10 codes (codelists openly available in Green[30]), primary diagnosis (diagnostic position 1) and for emergency (unplanned) admissions only. All ambulatory care sensitive conditions were selected as an overall proxy for avoidable hospitalisations. Ambulatory care sensitive conditions were defined as conditions that can be treated effectively in the community and should not therefore require hospital admission.[12 13 29] We further disaggregated all ambulatory care sensitive conditions into three measures representing the type of condition: (a) acute conditions (eg, cellulitis, dental caries, rickets, gastric ulcer), (b) chronic conditions (eg, hypertension, angina, asthma), (c) vaccine-preventable conditions (eg, mumps, measles, influenza). We did not include admissions due to COVID-19 as avoidable or vaccine-preventable conditions as we wanted to examine the indirect impacts of the pandemic (as well as ensuring a consistent set of conditions pre-pandemic). We also included emergency urgent care sensitive conditions as an alternative measure of avoidable hospitalisations. These are acute exacerbations of urgent conditions that will result in hospital admission but the NHS should be able to treat within the community to minimise the need for hospital care.[12 13 29] While emergency urgent care sensitive conditions overlap with some ambulatory care sensitive conditions, they also include certain conditions not included elsewhere (such as mental and behavioural conditions and falls). Finally, we also included a measure of all emergency hospital admissions to provide context for our measures.

## Measures of social inequality

We considered how trends in our outcome variables varied by three measures of social and spatial inequality that have been widely reported to have been associated with unequal COVID-19 outcomes.[20 23] Neighbourhood socioeconomic deprivation was measured using the Index of Multiple Deprivation 2019.[31] Individuals were matched to Lower Super Output Area by their home residence in each month and we calculated the quintile of deprivation rank. There were no alternative and more recent data available that could measure neighbourhood deprivation during the full period. Ethnicity was recorded in the electronic health record and we consider four groups (white or white British, black or black British, Asian or Asian British, and mixed ethnicity). We did not undertake further disaggregation because of issues with small numbers. We excluded individuals with 'other' ethnicity since the group is extremely heterogeneous, limiting our scope to draw conclusions (they represented 2.2% of the entire sample). Patients were allocated to seven government office regions, based on home residence, to capture regional differences. Missing data for covariates are presented in online supplemental table A.

## Statistical analyses

We estimated summary statistics for each outcome measure using aggregated counts from individual records. Where aggregated statistics were calculated, we used crude rates to measure monthly hospital admission rates (population was defined as total number of people alive on the first of each month within the OpenSAFELY-TPP data). Where we stratified measures by sociodemographic indicators, we estimated directly age-standardised rates to adjust for the different age structures of each group that may confound trends. All measures were stratified by sex (female or male). We visualised rates and presented descriptive statistics to investigate trends.

To quantify and measure the extent of how trends in inequalities have changed over time, we used a difference-in-differences regression model. Difference-in-differences is a commonly used method for evaluating natural experiments as it estimates the average change of time of an outcome following an event. This helps us to quantify how trends have changed since the start of the pandemic. Here we fit a separate linear regression model using the monthly observations for each sociodemographic measure independently as the dataset (ie, three separate models). The direct age-standardised admission rate of each month was used as our outcome variable, with a binary predictor variable for if the month was pre-pandemic or post-pandemic (defined here as March 2020 onwards), a categorical variable for the specific sociodemographic measure of inequality (deprivation quintile, ethnic group or geographic region), and an interaction effect between the binary and categorical variables. Models were stratified by sex. We focus on the interaction effect here since it tells us how the trend changed overall, indicating if inequalities between sociodemographic groups narrowed or widened. We excluded all missing data from analyses.

## Patient and public involvement

There was no public engagement in the design or completion of this project.

## RESULTS

### Overall population-level trends

Table 1 presents key summary statistics for our data. During the study period, there were 6 645 550 emergency hospital admissions, with a monthly average of 170 399. There were 1 129 770 ambulatory care sensitive hospital admissions (17% of all emergency admissions), chronic ambulatory care sensitive hospital admissions being the most common. There 1 031 205 emergency urgent care sensitive hospital admissions (16% of all emergency admissions).

Figure 1 presents crude admission rates by sex over the study period for each of our outcome measures. Trends in emergency hospital admissions were stable and consistent throughout 2019 (figure 1A). In 2020, there were sudden large falls in emergency admissions coinciding with the first national lockdown in England (eg, in April 2020 rates were 44% lower (females) and 39% lower (males) than compared with rates in January 2020). Emergency admission rates then increased quickly in the period following the national lockdown, although increases did not reach the same levels as in 2019. Rates fell once more at the end of 2020 (again coinciding with national lockdowns),

**Table 1** Frequency counts for outcome measures (1 January 2019 to 31 March 2022)

| Measure | Total over study period | Mean per month |
|---|---|---|
| Emergency admissions | 6 645 550 | 170 399 |
| All ambulatory care sensitive admissions | 1 129 770 | 28 968 |
| Acute ambulatory care sensitive admissions | 292 630 | 7503 |
| Chronic ambulatory care sensitive admissions | 563 740 | 14 455 |
| Vaccine-preventable ambulatory care sensitive admissions | 282 630 | 7247 |
| Emergency urgent care sensitive admissions | 1 031 205 | 26 441 |
| Population | 918 302 035 | 23 546 206 |

Overall trends in hospital admissions

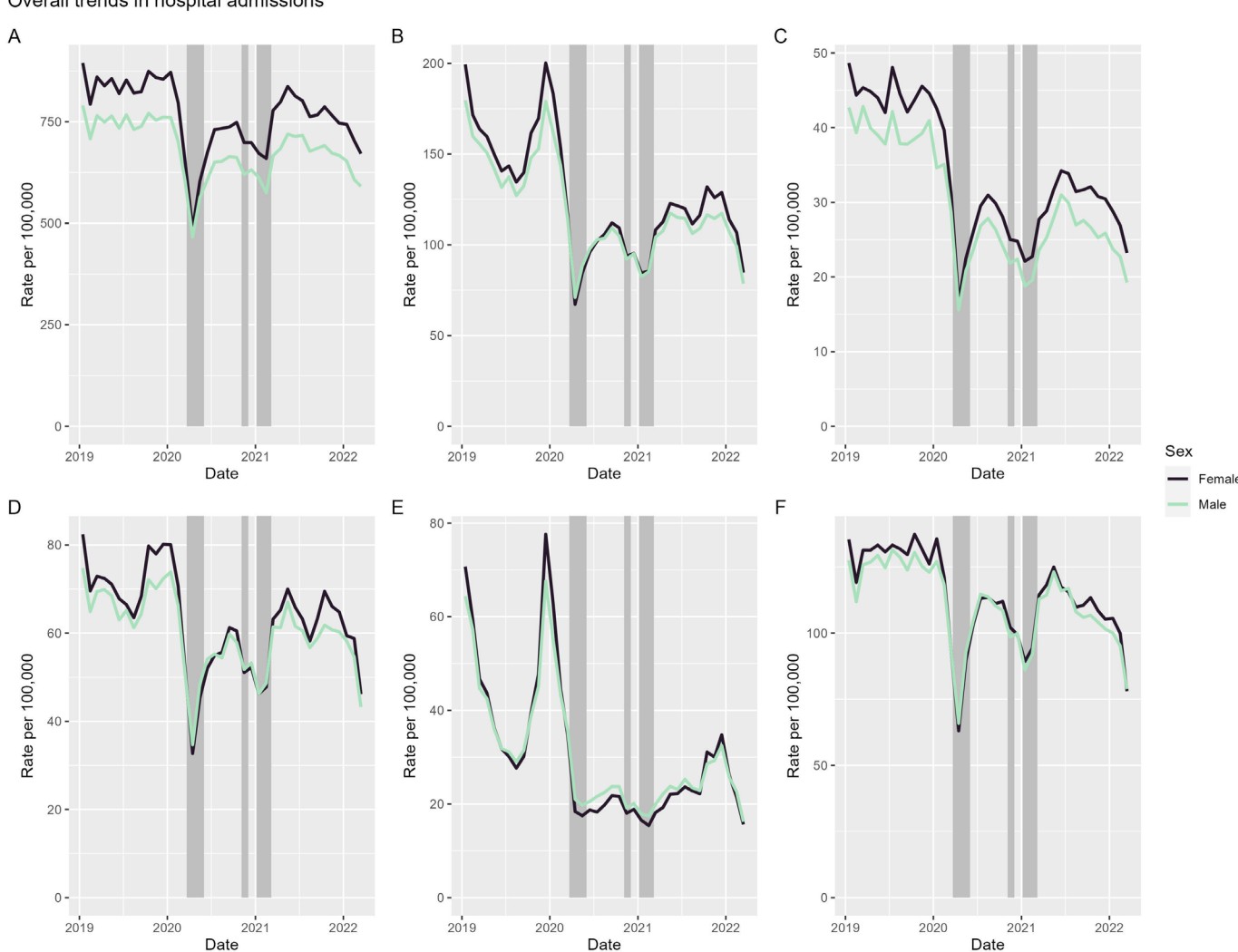

A = Total emergency hospital admissions, B = All ambulatory care sensitive admissions,
C = Acute ambulatory care sensitive admissions, D = Chronic ambulatory care sensitive admissions,
E = Vaccine-preventable ambulatory care sensitive admissions, F = Emergency urgent care sensitive conditions.

**Figure 1** Crude hospital admission rates (per 100 000 population) by sex for measures of avoidable hospitalisations. Shaded periods represent national lockdowns.

subsequently recovering quickly and remaining at a level higher in 2021 than in 2020, but still less than in 2019.

Patterns of avoidable hospitalisations largely follow similar trends to emergency hospital admissions. There were large falls in all, acute, and chronic ambulatory care sensitive hospital admissions and emergency urgent care sensitive admissions between March and May 2020. These falls were then followed by large rises, although not to 2019 levels, before subsequent falls during the second and third national lockdowns (with large rises following). Vaccine-preventable ambulatory care sensitive admissions were the only measure that did not follow this trend. There were distinct peaks in winter 2019 but, after December, rates fell sharply and then continued at a low level throughout all of 2020. Rates then began to rise throughout 2021 (although only to the lowest points of 2019), peaking in December 2021, before falling thereafter.

### Trends by deprivation
We next examined whether trends in our measures varied by deprivation. Online appendix figures A and B present trends for females and males, respectively. There were minimal differences by sex so we described the findings together. A distinct social gradient was evident for all outcomes, with the highest rates in the most deprived quintile and lowest in the least deprived quintile. Trends over the study period largely followed those reported at the population level and were consistent across deprivation quintiles.

Figure 2 presents the interaction effects from our difference-in-differences models. The associations are negative which suggest that inequalities by deprivation narrowed over the study period. Only for the most deprived quintile did the 95% CIs not cross a value of 0 for all outcomes. The extent of this narrowing of inequalities was not consistent throughout the study period. The

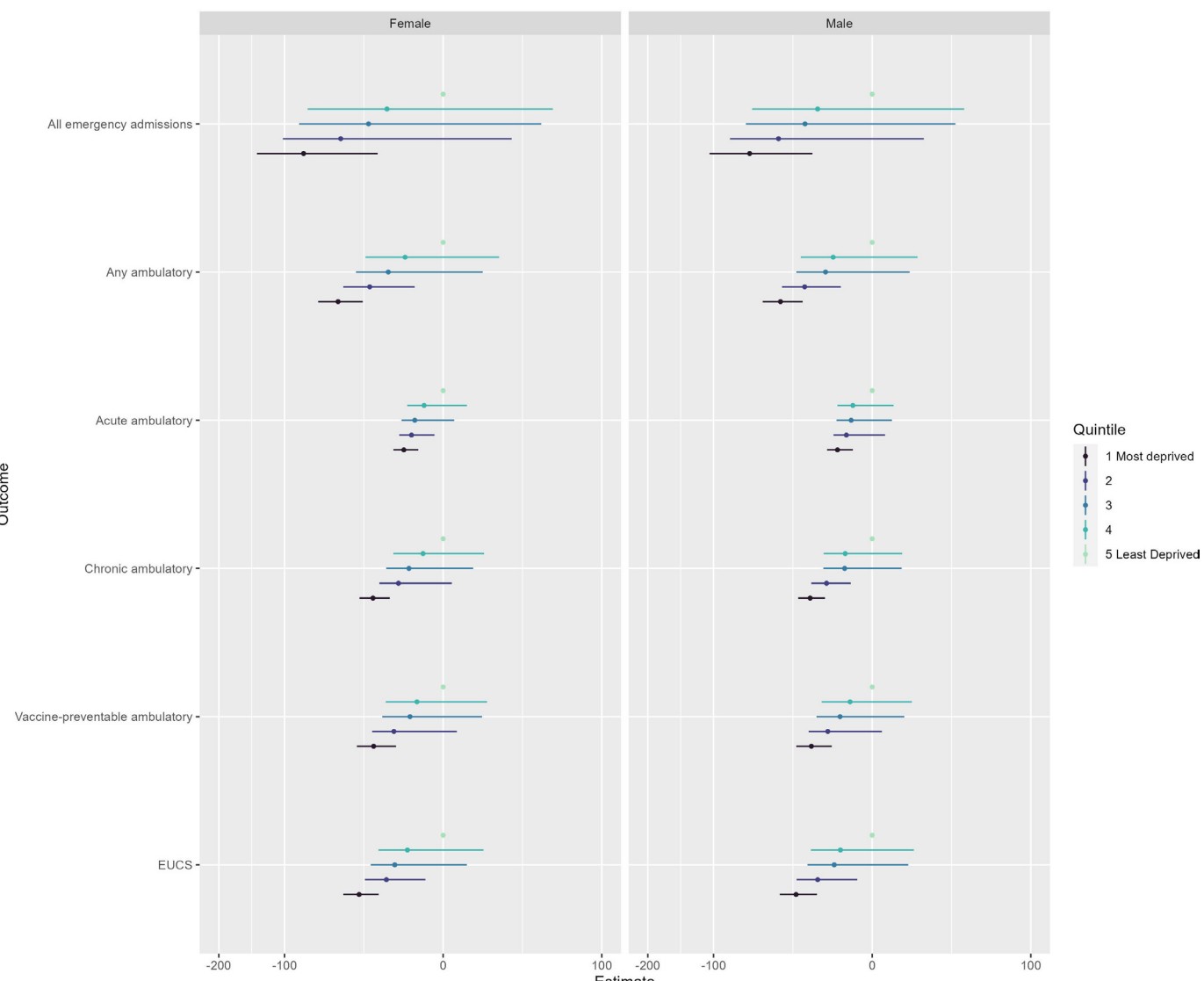

**Figure 2** Estimated interaction effects (with 95% CIs) between start of the pandemic and deprivation quintile from a series of difference-in-differences regression models for six outcome variables. Least deprived quintile was selected as the reference group. Ambulatory, ambulatory care sensitive admissions; EUCS, emergency urgent care sensitive admissions.

range between most and least deprived quintiles was smallest during the first national lockdown. For example, the range for females was 65 per 100 000 in April 2020 compared with 156 per 100 000 in January 2020. The range of values then increased following the lockdown but always remained smaller than what was observed in 2019. This would support the notion that absolute levels of social inequalities by deprivation have narrowed. However, relative differences remained similar (eg, 2.4 times higher in April 2020 compared with 2.3 times higher in January 2020 for females) suggesting that inequalities were still important determinants.

### Trends by ethnicity

Online supplemental figures C and D present trends by ethnic group and sex. The general trends for each outcome match those described in earlier sections, with the falls in hospital admissions during key COVID-19 waves consistent across all ethnic groups. While there

is some ordering of rates by ethnic group, the ordering itself is not consistent and CIs often overlap, suggesting that any ordering is not meaningful. There were no clear observable differences between groups during the main waves/lockdowns.

When estimating how trends changed overall (figure 3), we did not find evidence of significantly different trends for mixed and black ethnic groups in comparison to white ethnicity. For Asian ethnicity, we find evidence of negative associations which suggests that trends in avoidable hospitalisations declined at a greater rate than compared with the white ethnic group over the period.

### Trends by region

Online supplemental figures E and F present trends by geographical region for England. Trends largely follow those described previously, with falls in admissions across all outcomes during national lockdowns consistent across all regions. There were noticeable inequalities

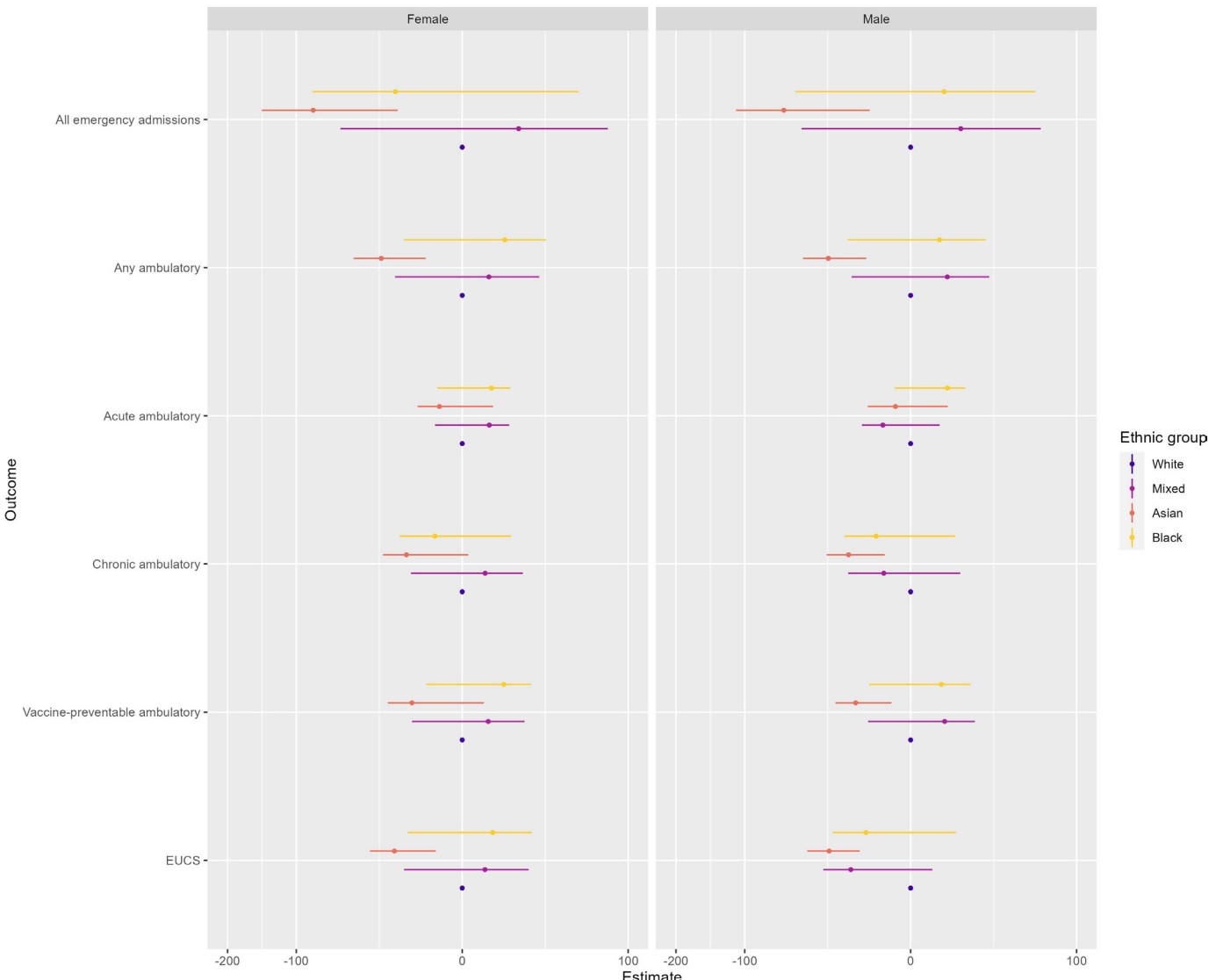

**Figure 3** Estimated interaction effects (with 95% CIs) between start of the pandemic and ethnic group from a series of difference-in-differences regression models for six outcome variables. White ethnicity was selected as the reference group. Ambulatory, ambulatory care sensitive conditions; EUCS, emergency urgent care sensitive.

between regions. Rates in the West Midlands were consistently highest for most outcomes (other than vaccine-preventable care sensitive admissions), followed by North West and North East. Admission rates in the South East and South West were consistently lowest across all periods and for all outcomes. London is the exception, lying in the middle for most outcomes. However, during the first national lockdown (and to a lesser extent at the end of 2020), falling rates take London to the lowest values.

When estimating the overall change in trends, we found inconsistent patterns (figure 4). Yorkshire and West Midlands had the greatest number of estimates that were statistically significant. Northern regions also saw negative associations in ambulatory care sensitive conditions for females, but not males (in comparison to the South East).

## DISCUSSION
### Key results
Using electronic health records for an average of 23 million people per month, we demonstrate that the COVID-19 pandemic led to a step-change in 'avoidable' hospital admissions. All measures of avoidable hospitalisations we analysed declined during the first national lockdown. While rates increased subsequently (other than in periods of national lockdowns), avoidable hospitalisations have remained lower than 2019 levels. Vaccine-preventable ambulatory care sensitive conditions have remained low for 2020 and 2021, and did not follow trends for other measures of avoidable hospitalisations. Avoidable hospitalisations were highest in the most deprived areas, Yorkshire and West Midlands, and inconsistent across ethnic groups. We find evidence of narrowing absolute levels of inequalities by deprivation, ethnicity and geographical

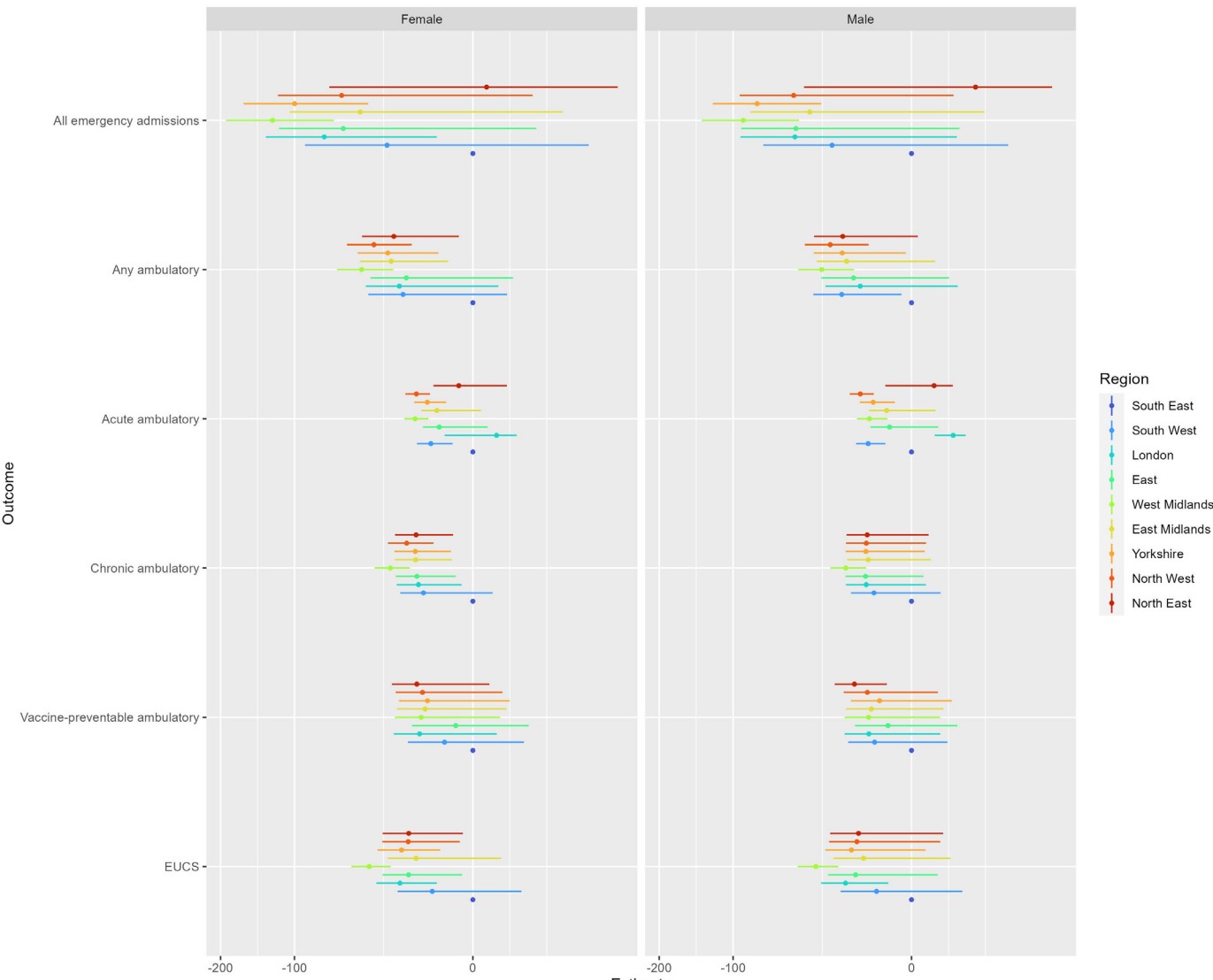

**Figure 4** Estimated interaction effects (with 95% CIs) between start of the pandemic and geographic office region from a series of difference-in-differences regression models for six outcome variables. South East region was selected as the reference group. Ambulatory, ambulatory care sensitive admissions; EUCS, emergency urgent care sensitive admissions.

region. The largest falls in inequalities occurred during the first national lockdown and have remained lower thereafter.

## Interpretation

Our study is one of the largest to investigate how avoidable hospitalisations changed before and during the COVID-19 pandemic. We find large falls in avoidable hospitalisations during periods of national lockdowns. During periods when society opened up, rates increased but never to 2019 levels suggesting a gradual recovery in population-level trends in avoidable hospitalisations. These findings follow similar trends observed in other countries despite their differences in health systems, sociocultural contexts and experiences of COVID-19.[15–18] We extend this work through analysing a longer time period and through detailed disaggregated by markers of socioeconomic inequality. Our results also follow

observations for other metrics including falls in non-COVID-19 mortality.[32 33]

While one interpretation might therefore be that the impacts of healthcare disruption were minimal, we should not forget that healthcare disruption is part of the explanation for the trends we observe. During the first national lockdown, the NHS cancelled or postponed or many patients planning to be admitted to free capacity for patients with COVID-19.[3 4 6] Individuals may also have avoided going to hospital if at all possible for fear of being exposed to SARS-CoV-2.[34] Such disruption may explain why we saw the largest decreases in rates during the first lockdown; the NHS was trying to avoid admitting people to hospital unless events were life threatening or required urgent care. More people dying at home rather than at hospital may have translated into fewer hospital admissions,[35] although one study found that people who died at home were more likely to have had an emergency

admission during the pandemic.[36] Deaths at home also consistently increased across all ethnic groups and levels of deprivation, suggesting that they could not explain the narrowing of inequalities.[36] The extent of declines in trends is likely to reflect significant changes in patient behaviour (eg, fear of being exposed to SARS-CoV-2 in healthcare settings, not wanting to burden services in a crisis) and access to health systems (eg, physicians deciding not to refer patients to hospital, changes in how admissions are made, prioritisation of community care). Unpicking the exact drivers is complex, especially in how they affected the different dimensions of inequalities, given the numerous interacting factors that may have produced this drop in avoidable hospitalisations and represents an important research gap.

Vaccine-preventable ambulatory care sensitive hospital admissions had markedly different trends to the other types of avoidable hospitalisations. Low levels of vaccine-preventable ambulatory care hospital admissions in 2020–2021 reflects the low level of influenza.[37] Influenza normally accounts for a large proportion of these hospital admissions but, as with all respiratory infections, of its transmission is reduced by the non-pharmaceutical interventions adopted for SARS-CoV-2.[38 39] Government advice on shielding and changes in behaviours during the pandemic may also explain these trends. Reductions in other viral and respiratory conditions, such as asthma exacerbations in 2020, also follow the same observed trends.[33] However, now that all COVID-19 related non-pharmaceutical interventions have been relaxed, there is a need to prepare for co-occurring waves of SARS-CoV-2 and influenza.[40]

The impacts of the COVID-19 pandemic have in many respects been highly unequal.[20] While our results provide evidence on the unequal distribution of avoidable hospitalisations across social and spatial characteristics, the changing trends in inequalities are not always so obvious, consistent and often nuanced in their interpretation. Contrary to our hypothesis, we find evidence that absolute levels of inequalities in avoidable hospitalisations across deprivation, ethnicity and region narrowed. Counterintuitively for deprivation, the narrowing of inequalities occurred during the first national lockdown, suggesting that the period of greatest disruption in the access to healthcare did not disproportionally impact people in these areas. We are not aware of any specific government interventions that would explain these trends. An evaluation of local and national policies that might have narrowed inequalities would represent important future research. We cannot discount entirely the possibility that this is a consequence of fewer hospital admissions during these periods. It may be that individuals who would have normally been captured in avoidable hospitalisations had experienced COVID-19, been hospitalised and died as a result instead (especially given the disproportionate distribution of unplanned hospitalisation and COVID-19 outcomes by our measures of inequality). However, we did not find evidence that inequalities returned back to 2019 levels during periods of low disruption or outside of COVID-19 waves which would suggest the driver is different.

Although inequalities narrowed, there were still differences between regions, ethnic groups and levels of deprivation reminding us that inequalities remain important. More deprived areas and northern regions had consistently higher avoidable hospitalisations throughout the whole study period. There was a lack of literature on ethnic inequalities in avoidable hospitalisations pre-pandemic,[12] and our work contributes to showing how intricate they can be when investigating trends. Trends between black and mixed ethnic groups to white ethnicity were not different, which is different to experiences in the USA.[16] Our findings reiterate the need for policy makers to focus on tackling the root structural causes of health. Health is socially determined.[20] We need to better understand how and why the social determinants of health influence the risk of avoidable hospitalisations to identify areas we can intervene. These need to include both social (eg, people in more deprived areas working multiple jobs which don't have time to see their GP and end up using more flexible emergency care instead) and spatial factors (eg, are there cultural or infrastructure reasons that explain why northern regions perform poorly). While we do not find clear and consistent inequalities for ethnicity, this should not detract from the structural injustices that minoritised ethnic groups face which has implications for their health.[21]

### Limitations

Our analyses are descriptive and are unable to identify the reasons behind what the trends we observe. We did not have access to data pre-2019 that could help to disentangle our findings from longer-term trends. The findings may be subject to ecological fallacy as we had no measure of individual experience of healthcare disruption. However, the magnitude of changes are clear enough to suggest that they are real at least at the population level. While our electronic health records provide large representative data on patients,[28] they lack the detail about the contexts of individual's lives that can be found in other data types (eg, longitudinal surveys).[10]

By using a proxy measure of health system performance, we are also unable to ascertain the extent that our findings do reflect actual disruption to healthcare. Its impacts will be complex, occurring over the short-term and long-term. It may be that our study is too early to detect the full impact of this disruption and it will be necessary to continually monitor trends in our outcomes. Additionally, there is some debate in the literature over how valid avoidable hospitalisations are as a proxy for health system performance.[12] Although our measures are used by NHS England for measuring health system performance,[12] other COVID-19 related outcomes have witnessed widening social inequalities which are different to our findings.[1 20 21] Regional and hospital differences in reporting patterns may also exist. Validating our chosen outcomes is necessary for understanding the importance of our findings.

We use coarse groupings for each of our measures of inequality. For example, we group individuals by broad ethnic groups that hide the diversity within each group (eg, Asian constitutes communities from very different backgrounds and heritages). This was partly to minimise statistical disclosure issues, due to the lower number of events per month in some groups. Redacted counts (<10) only affected the ethnicity results for vaccine-preventable ambulatory care sensitive conditions where 69 monthly values (17.6%) were redacted. 61 of these data points were for the 'mixed' ethnic group between April 2020 and August 2021. No other outcomes and exposures were affected. Future research should explore further the scale and nature of the extent of inequalities, involving describing intersectional characteristics across age, sex, ethnicity, deprivation and region. Ideally, this would be hypothesis driven to avoid the risk of spurious associations. Similarly, moving beyond describing geographical inequalities by region to identify how trends vary across smaller places can help to present more precise patterns and help to identify potential drivers.

## CONCLUSIONS

The COVID-19 pandemic has affected avoidable hospitalisations in unexpected and potentially surprising ways. Worries that the pandemic would see rising hospital admissions due to disruptions in accessing care do not appear to have materialised, at least so far. While social and spatial inequalities narrowed during the pandemic, recent widening regional inequalities present cause for concern and present the case for renewed focus among narratives of 'building back better' and 'levelling up'.

### Information governance

NHS England is the data controller for OpenSAFELY-TPP and OpenSAFELY-EMIS; EMIS and TPP are the data processors; all study authors using OpenSAFELY have the approval of NHS England. This implementation of OpenSAFELY is hosted within the EMIS and TPP environments which are accredited to the ISO 27001 information security standard and are NHS IG Toolkit compliant.[41]

Patient data has been pseudonymised for analysis and linkage using industry standard cryptographic hashing techniques; all pseudonymised datasets transmitted for linkage onto OpenSAFELY are encrypted; access to the platform is via a virtual private network connection, restricted to a small group of researchers; the researchers hold contracts with NHS England and only access the platform to initiate database queries and statistical models; all database activity is logged; only aggregate statistical outputs leave the platform environment following best practice for anonymisation of results such as statistical disclosure control for low cell counts.[42]

The OpenSAFELY research platform adheres to the obligations of the UK General Data Protection Regulation and the Data Protection Act 2018. In March 2020, the Secretary of State for Health and Social Care used powers under the UK Health Service (Control of Patient Information) Regulations 2002 to require organisations to process confidential patient information for the purposes of protecting public health, providing healthcare services to the public and monitoring and managing the COVID-19 outbreak and incidents of exposure; this sets aside the requirement for patient consent.[43] This was extended in July 2022 for the NHS England OpenSAFELY COVID-19 research platform.[44] In some cases of data sharing, the common law duty of confidence is met using, for example, patient consent or support from the Health Research Authority Confidentiality Advisory Group.[45]

Taken together, these provide the legal bases to link patient datasets on the OpenSAFELY platform. GP practices, from which the primary care data are obtained, are required to share relevant health information to support the public health response to the pandemic, and have been informed of the OpenSAFELY analytics platform. This study was classified as service evaluation.

### Data access

Access to the underlying identifiable and potentially re-identifiable pseudonymised electronic health record data is tightly governed by various legislative and regulatory frameworks, and restricted by best practice. The data in OpenSAFELY is drawn from General Practice data across England where EMIS and TPP are the data processors.

EMIS and TPP developers initiate an automated process to create pseudonymised records in the core OpenSAFELY database, which are copies of key structured data tables in the identifiable records. These pseudonymised records are linked onto key external data resources that have also been pseudonymised via SHA-512 one-way hashing of NHS numbers using a shared salt. Bennett Institute for Applied Data Science developers and PIs holding contracts with NHS England have access to the OpenSAFELY pseudonymised data tables as needed to develop the OpenSAFELY tools.

These tools in turn enable researchers with OpenSAFELY data access agreements to write and execute code for data management and data analysis without direct access to the underlying raw pseudonymised patient data, and to review the outputs of this code. All code for the full data management pipeline—from raw data to completed results for this analysis—and for the OpenSAFELY platform as a whole is available for review at github.com/OpenSAFELY.

**Author affiliations**
[1]Geography & Planning, University of Liverpool, Liverpool, UK
[2]European Centre on Health of Societ, London, UK
[3]Nuffield Department of Primary Care Health Sciences, Oxford University, Oxford, UK
[4]Medicines and Diagnostics Policy Unit, NHS England, London, UK
[5]University of Bristol, Bristol, UK
[6]Division of Community Health Sciences, University of Edinburgh, Edinburgh, UK
[7]The University of Edinburgh Usher Institute of Population Health Sciences and Informatics, Edinburgh, UK
[8]Oxford University, Oxford, UK
[9]UCL, London, UK
[10]MRC/CSO Social & Public Health Sciences Unit, University of Glasgow, Glasgow, UK

**Acknowledgements** We are very grateful for all the support received from TPP technical operations team throughout this work, and for generous assistance from the information governance and database teams at NHS England and the NHS England Transformation Directorate. Information about data sharing, data access and information governance can be found at the end of this document.

**Contributors** MAG, MM and SVK came up with the idea for the study. MAG, MM and SVK designed the study, with input from AS, JM and SAS. JM, BM, AM and SB were involved in data access and collection. MAG and JM manipulated data into an analysis ready format. MAG led on data analyses, supported by SVK. MAG, MM and SVK wrote the initial draft of the paper, with all other authors revising the paper. The data management and analysis code for this paper was led by MAG and contributed to by all named authors. MAG acts as the guarantor of the paper and accepts full responsibility for the work and/or the conduct of the study, had access to the data, and controlled the decision to publish. A full list of people in the 'Longitudinal Health and Wellbeing National Core Study Collaborative can be found at https://www.ucl.ac.uk/covid-19-longitudinal-health-wellbeing/longitudinal-health-and-wellbeing-collaborative. Members of the OpenSAFELY consortium can be viewed at https://www.opensafely.org/team/.

**Funding** This work was funded as part of the National Core Studies Longitudinal Health and Wellbeing programme (MC_PC_20030, MC_PC_20059). MAG acknowledges funding for this research from UK Medical Research Council (MR/W021242/1). SVK acknowledges funding from a NRS Senior Clinical Fellowship (SCAF/15/02), the Medical Research Council (MC_UU_00022/2) and the Scottish Government Chief Scientist Office (SPHSU17). JM is partly funded by the National Institute for Health and Care Research Applied Research Collaboration West (NIHR ARC West). The full statement, listing the names of all relevant NCS Consortium staff, can be found here: https://www.ucl.ac.uk/covid-19-longitudinal-health-wellbeing/convalescencestudy-collaborative.

**Competing interests** None declared.

**Patient and public involvement** Patients and/or the public were not involved in the design, or conduct, or reporting, or dissemination plans of this research.

**Patient consent for publication** Not applicable.

**Ethics approval** This study was supported by Professor Martin McKee (Honorary Consultant at UCLH NHS Foundation Trust) as senior sponsor, and approved by the University of Liverpool's Research Ethics Board (reference 10 634).

**Provenance and peer review** Not commissioned; externally peer reviewed.

**Data availability statement** Data may be obtained from a third party and are not publicly available. All data were linked, stored and analysed securely within the OpenSAFELY platform: https://opensafely.org/. Data include pseudonymised data such as coded diagnoses, medications and physiological parameters. No free text data are included. All code is shared openly for review and re-use under MIT open license on https://github.com/opensafely/avoidable_hospitalisations_trends. Detailed pseudonymised patient data is potentially re-identifiable and therefore not shared. Primary care records managed by the GP software provider, TPP, were linked to Hospital Episode Statistics through OpenSAFELY.

**ORCID iDs**
Mark Alan Green http://orcid.org/0000-0002-0942-6628
Martin McKee http://orcid.org/0000-0002-0121-9683
Jon Massey http://orcid.org/0000-0002-2497-4040
Syed Ahmar Shah http://orcid.org/0000-0001-5672-0443

Srinivasa Vittal Katikireddi http://orcid.org/0000-0001-6593-9092

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
