## [Reviewer comments · BMJ Open]

ARTICLE DETAILS

TITLE (PROVISIONAL)	Trends in inequalities in avoidable hospitalisations across the COVID-19 pandemic: A cohort study of 23.5 million people in England
AUTHORS	Green, Mark; McKee, Martin; Massey, Jon; Mackenna, Brian; Mehrkar, Amir; Bacon, Seb; Macleod, John; Sheikh, Aziz; Shah, Syed Ahmar; Consortium, The OpenSAFELY; National Core Study Collaborative, The Longitudinal Health and Wellbeing; Katikireddi, Srinivasa

VERSION 1 – REVIEW

REVIEWER	Wan, Yize Queen Mary University of London
REVIEW RETURNED	10-Aug-2023

GENERAL COMMENTS	The authors present an analysis using OpenSAFELY assessing changes in avoidable hospitalisations across periods of healthcare disruption from the COVID-19 pandemic. The study examines if differences also exist across groups of potential inequality by SE deprivation, ethnicity, and geographic region. The manuscript is well-presented, and the findings are interesting and add to existing literature. I have a few comments and general suggestions. Methods The difference between ambulatory care sensitive conditions and emergency urgent care sensitive conditions is slightly unclear (p4, lines 46-60). The definitions given give me the impression that the latter is a subgroup of the former which does not seem to fit how the subsequent findings are presented. Please could you further clarify this and explain these categories. Results The findings report different trends for vaccine-preventable ambulatory hospital admissions particular that it fell in 2019 then was at a low level in 2020 (p7, lines 38-44). It might be worth considering and discussing if some of this be related to government advice on shielding and changes in behaviour. Discussion I would also suggestion consideration of interactions between ethnicity, SE deprivation, and geographic region which may not be picked up in these analyses focusing on individual parameters. Although out of the scope of this study, as a potential for future directions it would also be Interesting and important to describe any changes during the post-pandemic recovery period with ongoing waiting lists at both GPs and hospitals.
--

REVIEWER	Van den Borre, Laura Vrije Univ Brussel, Sociology
REVIEW RETURNED	26-Sep-2023

GENERAL COMMENTS	Dear authors, Your manuscript is an important contribution to the field, as it provides a new perspective on socio-economic, ethnic and geographical inequalities in avoidable hospitalizations during the COVID-19 crisis. The study builds on high-quality data and sound methodology. The results are surprising and provide much food for thought on the underlying mechanisms related to avoidable hospitalizations, as well as general hospitalizations during the COVID-crisis. However, I believe there are some points that deserve more clarification and/or should be elaborated further. Please find more specific questions and suggestions below. Introduction: -Please check references 15-18: Statement on “many other countries”, but the references seem to refer only to the English context. Methods: -p. 4 line 32: check relevance reference 29 for this statement. -P. 4 lines 34-35: “Total counts per month were rounded to their nearest 5 to minimize patient disclosure risks where low counts were evident.” Please provide more specific context in your response to this review: were there many situations with <5 cases/month? For which outcomes? How may this affect your results? Did you perform any sensitivity analyses? In addition, please include a statement on the impact of this method on your results in the limitation section (p. 10 lines 50-51), with for example for which time periods and outcomes this may have an impact. -p 4 line 53-56: You did not include COVID-19 admissions. How were these defined: as admissions WITH or DUE to COVID-19? Can you provide more insights into the possibility that some admissions were misclassified, and thus not missing from your analyses? For example, a person with hypertension admitted to the hospital also turns out to be infected with COVID-19, are they included in the analyses or not? -p 5 line 13: “Individuals were matched to lower super output area by their home residence in each month” Definitely a strength to have timely information on the neighborhood deprivation. Do you see any trends in transitions to less deprived socio-economic neighborhoods during the pandemic? Do you have any information on the relationship between geographical distribution of hospitals and more deprived areas (e.g., are hospitals as easy to reach for residents of more deprived areas compared to residents of less deprived areas?) -p. 5 line 26 Please provide more rationale for the difference-in-difference regression model. -With regard to the difference-in-difference regression model: how did you build the models? For example, for the trends in deprivation, did you control for ethnicity and geographical region?
---

	Results:  -All figures: please try to be consistent with the phrasing in the text and in the figures. In addition, please order the labels consistently (not alphabetically, but in the order of table 1) to increase the readability. -Figures 2-4: why do “all admissions” show such large confidence intervals, even for the geographical region with a limited share of missings? Discussion:  -p. 9 lines 9-12: please be more specific with regard to previous findings on mortality at home. -p. 9 lines 12-14: please be more specific with regard to previous findings on the number of admitted care to the NHS. -p. 9 lines 32-33: please be more specific with regard to previous findings on the reductions in viral and respiratory conditions: for the whole period under investigation; or some specific time periods? -Overall, how might policy measures have impacted your findings? Did your government implement measures to combat social and geographical inequalities in hospitalizations (or health in general)? -Overall, I think the discussion section would benefit from a broader discussion of socio-economic, ethnic and geographical health inequalities, and their mechanisms. For example, you said in the method section that the trends prior to 2019 were relatively flat, but the reference does not include any information on trends in deprivation, ethnic health inequalities or geographical health inequalities. Was there already a decreasing trend in the prepandemic situation, perhaps? And what about findings about a widening of social inequalities in COVID-19 related outcomes? How do you reconcile your findings with other evidence suggesting that clear ethnic inequalities for some groups in the risks of testing positive for SARS-CoV-2, COVID-19-related hospitalizations, COVID-19-related ICU admission and mortality, even after accounting for differences in sociodemographic, clinical, and household characteristics (See your reference 9)? Further, in what way would the mechanisms behind avoidable hospitalization be different from COVID-19 hospitalization for your three key characteristics? Is there any evidence of socio-economic, ethnic and geographical differences in the risk of dying at home; or patient behaviors? Details:  -Typo in the strengths and weaknesses list: “it us uncertain” should be “it is uncertain” -p. 3 line 18: “also many countries internationally”; internationally feels redundant in the sentence
--	---

VERSION 1 – AUTHOR RESPONSE

Reviewer: 1	
The authors present an analysis using OpenSAFELY assessing changes in avoidable hospitalisations across periods of healthcare disruption from the COVID-19 pandemic. The study examines if differences	Thank you for your kind and positive comments. We appreciate the encouragement.

also exist across groups of potential inequality by SE deprivation, ethnicity, and geographic region. The manuscript is well-presented, and the findings are interesting and add to existing literature. I have a few comments and general suggestions.	
Methods	
The difference between ambulatory care sensitive conditions and emergency urgent care sensitive conditions is slightly unclear (p4, lines 46-60). The definitions given give me the impression that the latter is a subgroup of the former which does not seem to fit how the subsequent findings are presented. Please could you further clarify this and explain these categories.	To clarify, emergency urgent care sensitive conditions are not a subgroup of ambulatory care sensitive conditions. They contain different health conditions (ICD-10 codes) that are not necessarily always found in both. We have added some additional information in the methodology that clarifies how emergency urgent care sensitive conditions are differ: “While emergency urgent care sensitive conditions overlap with some ambulatory care sensitive conditions, they also include certain conditions not included elsewhere (such as mental and behavioural conditions and falls).” (page 5) We have left the exact definitions as they are since they follow the guidance from NHS England in how to define the measures.
Results	
The findings report different trends for vaccine-preventable ambulatory hospital admissions particular that it fell in 2019 then was at a low level in 2020 (p7, lines 38-44). It might be worth considering and discussing if some of this be related to government advice on shielding and changes in behaviour.	We agree with your comment, although with the qualification that it is probably impossible to distinguish between the two as, at least initially, public behaviour preceded government policy. We think this clarification belongs best in the discussion section and therefore we have added the following there: “Government advice on shielding and changes in behaviours during the pandemic may also explain these trends.” (page 9)
Discussion	
I would also suggestion consideration of interactions between ethnicity, SE deprivation, and geographic region which may not be picked up in these analyses focusing on individual parameters.	We cannot do this with the data, since we are analysing monthly counts stratified independently by each measure of inequalities. To do this would require us to re-process the data and extract new data (i.e., monthly counts for each combination of the interaction) which would be currently challenging due to ongoing changes in data governance. We think that this is not appropriate for our paper and outside its scope since it is not something that we pre-specified in the protocol that we submitted for data access or is outlined in our study aim (i.e., we did not have a clear rationale for why we would expect the interaction effects described to be

	meaningful). Also, given the complex nature of these relationships, with the likelihood of many non-linear associations, there is considerable risk of producing spurious associations. Consequently, we politely decline to add in this new analysis. We do, however, feel that there is some merit in your suggestion. We have added some text into the limitations section to suggest that this is something future research should be exploring: “Future research should explore further the scale and nature of the extent of inequalities, involving describing intersectional characteristics across age, sex, ethnicity, deprivation and region. Ideally, this would be hypothesis driven to avoid the risk of spurious associations.” (page 11) We mention intersectionality here, rather than interactions, as we believe that it brings a more theory driven design to investigating the combination of measures.
Although out of the scope of this study, as a potential for future directions it would also be interesting and important to describe any changes during the post-pandemic recovery period with ongoing waiting lists at both GPs and hospitals.	We agree that this would be interesting. However, we fear that it could be incredibly difficult to make any sense of what we would find given so many other things are happening, including industrial action by health workers and managerial responses to them, many of which have quite localised impacts. We thank you for the suggestion though and will consider exploring it further in our future research where we can give it the considerable attention it deserves.
Reviewer 2	
Your manuscript is an important contribution to the field, as it provides a new perspective on socio-economic, ethnic and geographical inequalities in avoidable hospitalizations during the COVID-19 crisis. The study builds on high-quality data and sound methodology. The results are surprising and provide much food for thought on the underlying mechanisms related to avoidable hospitalizations, as well as general hospitalizations during the COVID-crisis. However, I believe there are some points that deserve more clarification and/or should be elaborated further. Please find more specific questions and suggestions below.	We value your supportive comments on the quality of our paper and thank you for highlighting its strengths.
Introduction:	

Please check references 15-18: Statement on “many other countries”, but the references seem to refer only to the English context.	Apologies, there appeared to be some issues with the referencing software that meant the correct references were not aligned. Thank you for spotting this – indeed we have noticed that quite a lot of the references were misaligned and not correct. The individual in-text references we inserted were correct, however the reference list was erroneous producing a mismatch between the numbers. We have now revised and used the correct references throughout, with international literature being appropriately cited. Apologies for this oversight.
Methods:	
p. 4 line 32: check relevance reference 29 for this statement.	See above response.
P. 4 lines 34-35: “Total counts per month were rounded to their nearest 5 to minimize patient disclosure risks where low counts were evident.” Please provide more specific context in your response to this review: were there many situations with <5 cases/month? For which outcomes? How may this affect your results? Did you perform any sensitivity analyses? In addition, please include a statement on the impact of this method on your results in the limitation section (p. 10 lines 50-51), with for example for which time periods and outcomes this may have an impact.	This decision was part of the requirements for using OpenSAFELY data and was introduced by NHS England. It is part of their statistical disclosure controls. We were required to follow this. This was applied for all outcome measures. The rounding of counts to their nearest 5 is unlikely to have any significant impact on our findings. This is because the process only changed values (i.e., admission counts) by 1-3 units when rounding up/down. The difference, and its effects on rates, will therefore be minimal. For example, for the neighbourhood deprivation quintile outcomes, the total admitted persons per month (by sex) was often between 10,000 and 25,000 cases. Even in the least prevalent outcome – vaccine-preventable ambulatory care sensitive conditions – the monthly counts per IMD quintile were not lower than 300. Rounding up would have very little effect if any. We have added some additional clarity to the data section now: “Total counts per month for all outcomes were rounded to their nearest 5 to minimise risks of patient identification where low counts were evident (a condition in the standard data agreement). As this only adjusted monthly counts by 1-3 units, the impact was relatively minimal. We redacted counts <10 to meet statistical disclosure requirements.” (page 4) An analysis presented in the paper comparing their raw values to the ones we rounded would also not be possible by OpenSAFELY’s terms on statistical

	disclosure control. We were only approved to only output the rounded values. We redacted any counts <10. This did not affect any results for plots overall, or stratified by neighbourhood deprivation or region. The largest impact was for ethnicity measures and only occurred in the vaccine-preventable ambulatory care sensitive conditions data. Here, 69 data points (17.6%) were redacted. 61 of these points were for mixed ethnicity. Our study does not draw any specific conclusions about the mixed ethnicity group. We have now added this into the limitations section as you suggest: “Redacted counts (<10) only affected the ethnicity results for vaccine-preventable ambulatory care sensitive conditions where 69 monthly values (17.6%) were redacted. 61 of these data points were for the ‘mixed’ ethnic group between April 2020 and August 2021. No other outcomes and exposures were affected.” (page 11) Redacted data are not plotted on any of the graphs in the appendix.
p 4 line 53-56: You did not include COVID-19 admissions. How were these defined: as admissions WITH or DUE to COVID-19? Can you provide more insights into the possibility that some admissions were misclassified, and thus not missing from your analyses? For example, a person with hypertension admitted to the hospital also turns out to be infected with COVID-19, are they included in the analyses or not?	It was ‘due to COVID-19’. We have edited the text on page 4 to clarify this. We define each admission based on the primary admission (see page 4) as this is the main reason that an individual was hospitalised. This helps to avoid misclassification and is accepted practice to classify health records when measuring avoidable hospitalisations. Using secondary causes makes things messier as they are inconsistently applied and therefore may lead to greater misclassification. Since our measures of avoidable hospitalisations might also appear in these secondary positions, focusing on just the primary position helps make our data clearer to understand and interpret. We cannot rule out that COVID-19 was not present (measured or not in any of the secondary positions), but being placed in the secondary positions means it was not the primary reason a person was hospitalised. We think that this is a clear way of defining our outcomes and easier to interpret.
p 5 line 13: “Individuals were matched to lower super output area by their home residence in each month” Definitely a strength to have timely information on the neighborhood deprivation. Do you see any trends in transitions to less deprived	We did not have access to more granular neighbourhood deprivation measures that would allow us to measure the trends, and indeed, given restrictions on movement, coupled with low levels of digital access during the pandemic, it is not clear how this could have been obtained. This is

socio-economic neighborhoods during the pandemic? Do you have any information on the relationship between geographical distribution of hospitals and more deprived areas (e.g., are hospitals as easy to reach for residents of more deprived areas compared to residents of less deprived areas?)	a recognised limitation of available data in the UK (although even if we had more disaggregated data there would be questions about validity given small numbers and population movement). It certainly would be interesting to have in the analysis though. We have made a note of this limitation in the paper now: “There were no alternative and more recent data available that could measure neighbourhood deprivation during the full period.” (page 5) The relationship between hospital access and deprivation is not necessarily so clear. Many hospitals are located in urban areas which often are where concentrated deprivation exists – but not always. We agree that this might be an interesting issue to look at (i.e., how access intersects with deprivation and how this influences hospital admissions). This is especially the case within the context of the ‘inverse care law’ (that availability of health care is often located in areas that have the lowest need for them). However, this is beyond the scope of our paper and it would be difficult to do it justice within the tight word count that we are trying to remain within. We politely decline to add it in given that we have to keep under the word count.
p. 5 line 26 Please provide more rationale for the difference-in-difference regression model. With regard to the difference-in-difference regression model: how did you build the models? For example, for the trends in deprivation, did you control for ethnicity and geographical region?	We have added a brief statement as to why difference-in-differences regression was used: “Difference-in-differences is a commonly used method for evaluating natural experiments as it estimates the average change of time of an outcome following an event. This helps us to quantify how trends have changed since the start of the pandemic.” (page 5) We have revised the description of how the models were fit to make this clearer. This now reads: “Here we fit a separate linear regression model using the monthly observations for each socio-demographic measure independently as the dataset (i.e., three separate models). The directly age-standardised admission rate of each month was used as our outcome variable, with a binary predictor variable for if the month was pre- or post-pandemic (defined here as March 2020 onwards), a categorical variable for the specific socio-demographic measure of inequality (deprivation quintile, ethnic group or geographic region), and an interaction effect between the binary and categorical

	variables. Models were stratified by sex. We focus on the interaction effect here since it tells us how the trend changed overall, indicating if inequalities between socio-demographic groups narrowed or widened. We excluded all missing data from analyses.” (pages 5-6) We do not control for each factor in each model, since the monthly data are calculated for each socio-demographic variable separately. Hopefully this is clear in the above now. A suggestion from reviewer 1 to use interaction effects (see response in reviewer 1’s fourth comment/row for a longer response to them). In sum, this is not possible in our study since we only have data stratified independently by the socio-demographic measures. We have added the following text into the limitations section to note that interaction effects might be useful: “Future research should explore further the scale and nature of the extent of inequalities, involving describing intersectional characteristics across age, sex, ethnicity, deprivation and region. Ideally, this would be hypothesis driven to avoid the risk of spurious associations.” (page 11)
Results:	
All figures: please try to be consistent with the phrasing in the text and in the figures. In addition, please order the labels consistently (not alphabetically, but in the order of table 1) to increase the readability.	The terminology used in the text and all outputs (including both main paper and appendix) have been reviewed for their consistency, with relevant changes made throughout (see pages 6-8). Thank you for noticing this and the suggestion. We have reordered Figures 2-4 to match the order presented in Table 1. All outputs now are similar and follow your suggestions.
Figures 2-4: why do “all admissions” show such large confidence intervals, even for the geographical region with a limited share of missings?	Figures 2-4 plot the coefficients from the difference-in-difference regression models. The uncertainty is therefore reflective of the effect size. Where there are wide intervals, this is because there is large uncertainty in the direction and magnitude of the effect size. The regression models are estimated using the monthly rate values. As such, the wide confidence intervals merely represent the uncertainty in whether inequalities narrowed or not in the monthly time series data for our study period.
Discussion:	
p. 9 lines 9-12: please be more specific with regard to previous findings on mortality at home.	We have revised the statement to add further clarity. It now reads:

	“More people dying at home rather than at hospital may have translated into fewer hospital admissions (35), although one study found that people who died at home were more likely to have had an emergency admission during the pandemic (36).” (page 9)
p. 9 lines 12-14: please be more specific with regard to previous findings on the number of admitted care to the NHS.	We have removed this statement now as it was a speculative comment rather than evidence informed.
p. 9 lines 32-33: please be more specific with regard to previous findings on the reductions in viral and respiratory conditions: for the whole period under investigation; or some specific time periods?	We have now clarified the exact time period of the study we have cited to help the comparison that we make. It now reads as: “Reductions in other viral and respiratory conditions, such as asthma exacerbations in 2020, also follow the same observed trends (33).” (pages 9-10)
Overall, how might policy measures have impacted your findings? Did your government implement measures to combat social and geographical inequalities in hospitalizations (or health in general)?	We are not aware of any specific policy that might explain our findings – this does not discount that there might have been though. The ongoing COVID-19 Inquiry is evaluating government policies relevant to our observed trends and it has already revealed a large gulf between rhetoric and reality. Thus, this suggestion would be beyond the scope of our descriptive paper and require a bigger study to evaluate. This can be complicated when extending it to understanding geographical inequalities as local government policy decisions add an extra layer of decision making that makes it complicated to tease out what is happening. Based on your suggestion, we have added the following text: “We are not aware of any specific government interventions that would explain these trends. An evaluation of local and national policies that might have narrowed inequalities would represent important future research” (page 10) We also note that we refer to specific policies at various points during the discussion including:  - Added new text on importance of shielding recommendations in driving trends in vaccine-preventable ambulatory care trends (page 9). - Mention of impact of COVID-19 related non-pharmaceutical interventions in explaining trends in vaccine-preventable ambulatory care trends (page 10).
Overall, I think the discussion section would benefit from a broader discussion of socio-economic, ethnic and geographical health inequalities, and their mechanisms. For example, you said in the method section that	While we agree with your general comment, one could easily write a thesis on each of the three areas and their driving mechanisms. Our paper is currently 18 words under the word count (3982 out of 4000). Therefore, we would like to politely

the trends prior to 2019 were relatively flat, but the reference does not include any information on trends in deprivation, ethnic health inequalities or geographical health inequalities. Was there already a decreasing trend in the pre-pandemic situation, perhaps? And what about findings about a widening of social inequalities in COVID-19 related outcomes? How do you reconcile your findings with other evidence suggesting that clear ethnic inequalities for some groups in the risks of testing positive for SARS-CoV-2, COVID-19-related hospitalizations, COVID-19-related ICU admission and mortality, even after accounting for differences in sociodemographic, clinical, and household characteristics (See your reference 9)? Further, what way would the mechanisms behind avoidable hospitalization be different from COVID-19 hospitalization for your three key characteristics? Is there any evidence of socio-economic, ethnic and geographical differences in the risk of dying at home; or patient behaviors?

decline addressing the paper accordingly as we simply do not have the space to respond in sufficient detail on all the possible mechanisms that would fit in here.

We have clarified that the trends prior to 2019 reference was for “overall trends”. You are correct that the source does not look into inequalities – we found it difficult to find a good reference that could show that since 2010 inequalities across our three measures have changed or not. We know that inequalities by deprivation were flat 2010-2013 in this document (<https://www.health.org.uk/publications/qualitywatch-focus-on-preventable-admissions>), however this leaves out the period 2013-2019. We did not want to be misleading so leave it as “overall trends” (deleted text on page 9).

We did not have access to data before 2019 in OpenSAFELY so we are unable to assess trends pre-2019. We now have added this into the limitations section as well: “We did not have access to data pre-2019 that could help to disentangle our findings from longer-term trends.” (page 10).

As for the comment about how mechanisms differ between avoidable and COVID-19 hospitalisations, this is a complex issue to tease apart across our three measures of inequality. Such an endeavour is beyond the scope of our paper (since our aim is to just describe trends in avoidable hospitalisations) and therefore we politely decline to add it in. We would add that this is the same for explanations for why inequalities widened by ethnicity for COVID-19 outcomes – outside the remit of our paper. This does not discount that your comments were not valid or important, but that with a limited word count we have to be selective in what we can cover and therefore have chosen to focus just on the most relevant issues.

We have added additional evidence as to whether differences in patients dying at home might explain the level of inequalities:

“Deaths at home also consistently increased across all ethnic groups and levels of deprivation, suggesting that they could not explain the narrowing of inequalities (36).” (page 9)

	The report cited provides a source for the reader to explore these issues further. As for changes in patient behaviours varying by each of our measures during the pandemic, this is complex and there is not necessarily a clear answer. We have revised part of our discussion to highlight that this is a important research gap now: “Unpicking the exact drivers is complex, especially in how they affected the different dimensions of inequalities, given the numerous interacting factors that may have produced this drop in avoidable hospitalisations and represents an important research gap.” (page 9)
Details:	
Typo in the strengths and weaknesses list: “it us uncertain” should be “it is uncertain”	Thank you for spotting, this has been corrected (see page 2).
p. 3 line 18: “also many countries internationally”; internationally feels redundant in the sentence	We agree and this has been deleted (see page 3).